# A systemic approach for modeling soil functions

Hans-Jörg Vogel [1,5], Stephan Bartke [1], Katrin Daedlow [2], Katharina Helming [2], Ingrid Kögel-Knabner[3], Birgit Lang[4], Eva Rabot[1], David Russell[4], Bastian Stößel [1], Ulrich Weller[1], Martin Wiesmeier[3], and Ute Wollschläger[1]

[1]Helmholtz Centre for Environmental Research - UFZ, Permoserstr. 15, 04318 Leipzig, Germany
[2]Leibniz Centre for Agricultural Landscape Research (ZALF). Eberswalder Straße 84, 15374 Müncheberg, Germany
[3]TUM School of Life Sciences Weihenstephan, Technical University of Munich, Emil-Ramann-Straße 2, 85354 Freising, Germany
[4]Senckenberg Museum of Natural History, Sonnenplan 7, 02826 Görlitz, Germany
[5]Martin-Luther-University Halle-Wittenberg, Institute of Soil Science and Plant Nutrition, Von-Seckendorff-Platz 3, 06120 Halle/Saale, Germany

*Correspondence to:* Hans-Jörg Vogel (hjvogel@ufz.de)

**Abstract.** The central importance of soil for the functioning of terrestrial systems is increasingly recognized. Critically relevant for water quality, climate control, nutrient cycling and biodiversity, soil provides more functions than just the basis for agricultural production. Nowadays, soil is increasingly under pressure as a limited resource for the production of food, energy and raw materials. This has led to an increasing demand for concepts assessing soil functions so that they can be adequately
considered in decision making aimed at sustainable soil management. The various soil science disciplines have progressively developed highly sophisticated methods to explore the multitude of physical, chemical and biological processes in soil. It is not obvious, however, how the steadily improving insight into soil processes may contribute to the evaluation of soil functions. Here we present to a new systemic modeling framework that allows for a consistent coupling between reductionist yet observable indicators for soil functions with detailed process understanding. It is based on the mechanistic relationships between soil
functional attributes, each explained by a network of interacting processes as derived from scientific evidence. The non-linear character of these interactions produces stability and resilience of soil with respect to functional characteristics. We anticipate that this new conceptional framework will integrate the various soil science disciplines and help identify important future research questions at the interface between disciplines. It allows the overwhelming complexity of soil systems to be adequately coped with and paves the way for steadily improving our capability to assess soil functions based on scientific understanding.

## 1  Introduction

In 2015 the Food and Agriculture Organization of the United Nations (FAO) decreed the International Year of Soils, and the International Union of Soil Science initiated the International Decade of Soils (2015-2024). With these initiatives, the awareness of soil as a limited but essential resource has gained significant momentum through a considerable number of special events and publications. Undoubtedly, this is highly necessary given the enormous loss of soil through desertification
and degradation to the order of 12 million hectares per year (Noel, 2016), the conversion of vegetated land to building areas

notwithstanding. While today, climate, water and biodiversity is well perceived to be highly significant for life on earth, a comparable awareness with respect to soil has just begun to develop. It had been on the agenda since the Dust Bowl drought in the U.S. when the Soil Conservation Act was launched ("The history of every Nation is eventually written in the way in which it cares for its soil": F.D. Roosevelt), but this recognition has substantially declined since then.

In this paper, we do not reiterate the facts about the importance of the multitude of soil functions. This has been done in many recent publications at least partly triggered by the International Year of Soils (Amundson et al., 2015; Montanarella, 2015; Paustian et al., 2016; Keesstra et al., 2016; Adhikari and Hartemink, 2016). Instead, we focus on a key question related to ongoing research, which is far less addressed in the actual discussion on the importance of soil: What could the contribution of the soil sciences be to sustainable soil management? Here and in the remainder of the paper the term soil sciences is focused

on natural soil sciences including the classical disciplines of soil biology, soil chemistry and soil physics.

Recently, Keesstra et al. (2016) and Bouma and Montanarella (2016) addressed the question how soil scientists can help to reach the recently adopted UN Sustainable Development Goals (SDGs) in the most effective manner. They stress the key position of soil scientists within the stakeholder-policy arena in the role of an honest broker and the need to explicitly demonstrate and efficiently communicate the importance of soil in reaching the SDGs. Based on these and many similar publications, the

impression may be reached that our scientific knowledge on soil processes and how they produce emergent soil functions is pretty much settled and only insufficient is how to translate this knowledge into sustainable management practices. We are convinced that this is a misimpression — certainly not intended by the authors above. We do not question that a significant effort is required regarding knowledge transfer and implementation and that a transdisciplinary approach is highly required (Bouma, 2017, 2018). However we stress the fact that our knowledge on soil processes is fragmented throughout various disciplines and

the system perspective required to truly capture the reaction of soils to external forcing through land use and climate change is still in its infancy. This systemic approach is furthermore necessary considering the need to distinguish the enormous variety of different soil types in various geographic and climatic regions, all of whose functioning react specifically in response to external forcing.

Thus, what are the crucial research questions today? Adewopo et al. (2014) organized a poll among experts to determine the

priority research questions in soil science. Such an approach, however, bears the drawback that each expert cultivates his/her particular field of research. It certainly provides an excellent overview of the various research fronts, but the individual bias ultimately hampers a system perspective on soil processes, which we believe is highly needed. Another approach is to start from major societal concerns and how the soil sciences may contribute to corresponding solutions, as proposed by Baveye (2015). With respect to soils, there are mainly two major concerns: food security and the functioning of terrestrial systems.

Both are jeopardized by land use and climate change, having a direct impact on the contribution of soil functions to ecosystem services: food and fiber production, nutrient provisioning and cycling, climate regulation and carbon storage, water provision and quality maintenance, pollutant degradation and pest control, conservation of biodiversity. We still do not yet have a clear idea how to specifically measure all these functions (Baveye et al., 2016) and how these functions are related to the multitude of physical, chemical and biological processes interacting in soils. Indeed, this defines a formidable challenge for soil research,

calling for a systemic approach connecting the fragmented disciplines within the soil science community.

Such a system approach, providing a clear perspective on how soil functions emerge from small-scale process interactions, is a prerequisite to actually understanding the basic controls and to developing science-based strategies towards sustainable soil management. This will also have an enormous potential for facilitating communication towards stakeholders and policy makers by replacing the cacophony generated by a disciplinarily fragmented research community with harmonized information on the soil system's behavior.

In this contribution, we begin from societal demand and the manner in which socio-economic and soil systems are coupled. This defines the quality of information required for the communication between these two complex systems. Based on this, we then derive the actual challenges for soil research in light of the recognised ultimate goal of quantifying and predicting the impact of external forcing on the ensemble of soil functions. Finally, this leads to the proposal of a general framework for modelling soils as complex adaptive systems, thereby integrating the considerable amount of new insights on soil processes generated within the various disciplines of soil science during the last decades.

## 2    The human-soil interface

The general interaction and feedback loop between the socio-economic system and soil is depicted in Fig. 1. The impact of human activities on soils is induced by soil management in a wide range of habitat types from near-pristine landscapes through forests and grassland to agricultural land use. We only consider vegetated soil with a special focus on agriculture, such that "land use" might be reduced to "agricultural soil management". While the impact of soil management on soil properties and functions is evident – though still far from being understood in quantitative terms – the feedback from soils to socio-economic systems is less evident. Ultimately, it is brought about by the set of various soil functions that affect human resources. When looking at the current major societal challenges — food security and the functioning of terrestrial ecosystems — the soil functions listed in Fig. 1 need to be addressed. This is in accordance with European Commission (2006), but limited here to those soil processes related to the functioning of terrestrial ecosystems. The general feedback loop illustrated in Fig. 1 reflects the DPSIR framework (Smeets et al., 1999; Tscherning et al., 2012). Our main focus here is on the interface between the natural and the socio-economic system, which are soil management as driven by the latter and soil functions provided by the former.

Thus, there are well-definable links between the two systems, while both are internally highly complex. Within the socio-economic box, the challenge is to assess soil functions by some form of valuation system. This is increasingly discussed in the framework of ecosystem services. While various approaches of valuation are still a matter of debate (Baveye et al., 2016; Adhikari and Hartemink, 2016; Stolte et al., 2015; Robinson et al., 2014; Müller and Burkhard, 2012), the need of such a concept appears to be obvious. It is necessary to include the expected impact of soil management on soil functions in sustainability assessment and decision making. We do not reiterate the concept of soil ecosystem services; however, when asking for the contribution of soil science to the understanding of soil functions, we believe it is important to separate soil functions from soil ecosystem services and not to consider these terms to be synonymous, as explicitly proposed recently by Schwilch et al. (2016) or implied by Stavi et al. (2016). Soil functions are produced by complex interactions of natural processes

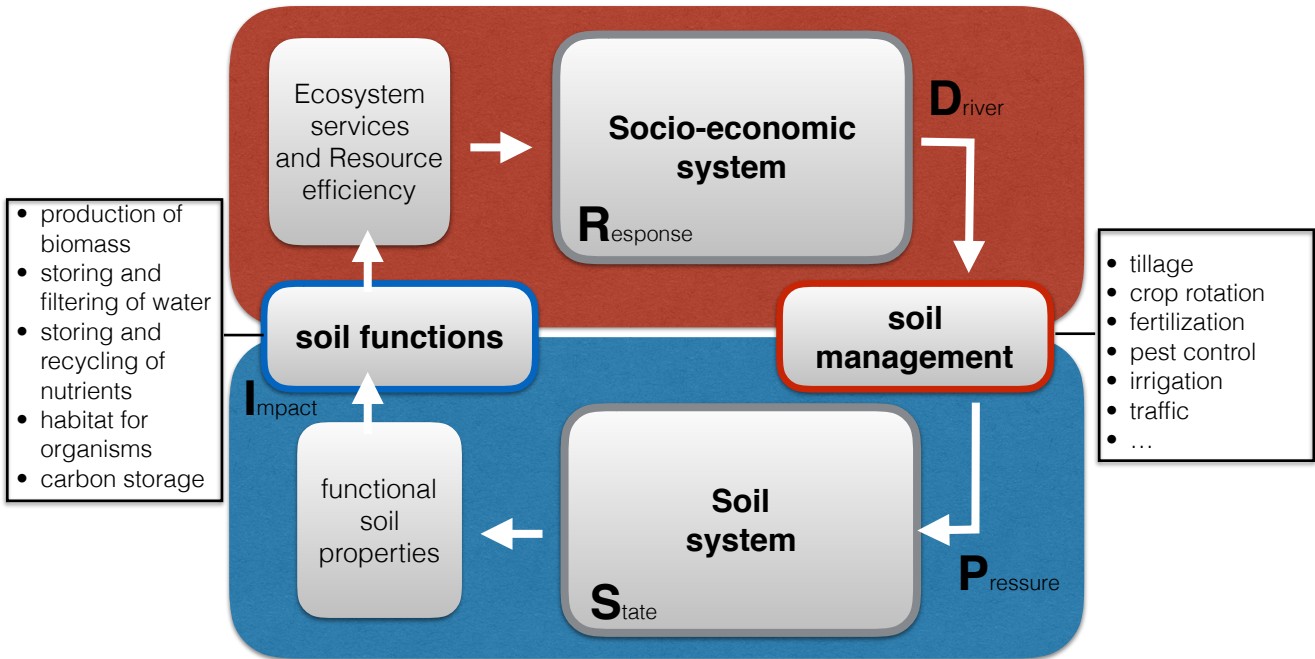

**Figure 1.** The human-soil interface related to the DPSIR framework. The obvious interfaces between the human and soil systems are soil management and soil functions. Human societies are the drivers of soil management producing various pressures on soils. A change in the soil's functionality may in turn provoke some response in soil management methods. Climate is another crucial driver for the soil system which is obviously required when zooming into the soil system (see Figs. 2 and 3).

and are the basis for the functioning of terrestrial ecosystems (located in the blue box in Fig. 1 without direct anthropogenic involvement), while ecosystem services and resource efficiency are defined in the context of the current human perception and may change according to the societal context (Spangenberg et al., 2014; Hauck et al., 2013).

Within the soil box of Fig. 1, the impact of pressures generated by soil management on the multitude of soil functions needs to be evaluated and predicted. For assessing management effects, substantial knowledge on the interaction between physical, chemical and biological properties driving processes in soil is required. In a following step, the set of soil functions needs to be derived from the ensemble of observable soil properties, so that the feedback loop can be closed. These soil functions are considered to emerge from the underlying processes, which are the core subject of soil research.

The general framework as illustrated in Fig. 1 is appealing in its simplicity. However, we recognize critical obstacles in the interplay between basic soil sciences and the social sciences needed to make such a framework operational. Those working on ecosystem services do not delve into the jungle of detailed soil processes. They typically stop at a quantification or estimation of soil functions using some proposed indicators (Dominati et al., 2014; Rutgers et al., 2012). There is little effort – since putatively not required – to go into greater depth about which underlying processes actually produce these functions.

On the other hand, soil scientists working on a detailed process understanding are scattered across different disciplines with limited cooperation among each other and with little awareness about the knowledge needs of the colleagues in social sciences. There has been enormous progress within these disciplines during the last decades, and a considerable arsenal of new methods for studying physical, chemical and biological processes in great detail is presently available. However, when it comes to a comprehensive understanding of emergent soil functions, the required systemic integration is still lacking. Thus, a gap exists where detailed process understanding needs to be converted to soil functions. Symptomatically, at the interface between the soil-science and socio-economic perspectives, the terminology becomes vague, hampering the communication even more, as recently noted by Schwilch et al. (2016). The good news is that the link between social sciences and natural sciences can be very clearly defined, as illustrated in Fig. 1; nonetheless, the interface between the two perspectives needs to be further developed. Based on this rough analysis, some crucial challenges for soil research are deduced in the following section.

## 3 The challenges for soil research

As stated in the previous section, the ultimate goal of soil research's support of sustainable soil management is to quantify and predict the impact of external forcing (right side in Fig. 1) on the ensemble of soil functions (left side in Fig. 1). Depending on local soil properties, this impact may range from ameliorative to destructive. The way soils react to the imposed forcing depends on a multitude of interacting physical, chemical and biological processes, and each soil function is considered to be an integrative property emerging from these interacting processes (Kibblewhite et al., 2008; Karlen et al., 2003).

A fundamental problem when analyzing the behavior of soil systems is that soil processes and their interactions are far too complex to be disentangled at the level of detailed individual processes – which in fact are very well understood in many cases – and then to rebuild the system behavior by combining all the individual processes. It is not the required computing power, which hinders such a bottom-up approach, but rather the lack of required information on soil properties, including their spatial heterogeneity and, most importantly, the highly non-linear character and multitude of process interactions.

A possible solution to this problem, which has been followed for quite some time, is the search for suitable indicators of soil health or individual soil functions, as already discussed in the previous section (Dominati et al., 2014; Rutgers et al., 2012; Moebius-Clune, 2016). Such indicators are based on observable soil properties that ideally reflect and integrate the variety of processes and their complex interactions at a higher level in a meaningful way. Thus, they are thought to contain sufficient information about these processes so that they can be used as proxies for quantifying soil functions. It should be noted that such indicators need to be evaluated in a site-specific way, since different soil types developed under different site conditions (i.e. geology, climate, relief, vegetation) behave differently, which is often ignored.

A prominent example is soil organic carbon (SOC, "humus") as an indicator for soil fertility or, more generally, for soil health (Franzluebbers, 2002; Loveland and Webb, 2003; Ogle and Paustian, 2005). This is because SOC has been recognized as supporting the stability of soil structure and thereby soil hydraulic properties and the physical habitat for soil organisms and their activity. Likewise, soil functions can be addressed from a physical perspective, i.e. soil structure being evaluated by an index based on observable hydraulic properties. This was suggested by Dexter (2004), who emphasized the close feedback

between soil structure, root growth, biological activity and, again, SOC. Biological indicators in the past stressed management effects on biodiversity, e.g. were conservation oriented, but recent developments emphasize methods indicating soil functions and general soil health (Doran and Zeiss, 2000; Kibblewhite et al., 2008; Ritz et al., 2009; Rutgers et al., 2012). These different approaches reflect the obvious interrelations between physical, chemical and biological agents in the soil systems.

The concept of using such indicators to estimate the state of soil in terms of soil functions is well justified and supported by substantial empirical evidence. However, if the impact of soil management is to be evaluated or measures for improving soil functions are to be developed, we need to focus on the dynamics of soil functions, i.e. their management-induced changes. This requires a profound understanding of the underlying processes and their interactions. For example, the soil's potential to store carbon is not just a simple measure of the capacity of some storage pool in soil; it depends on the type of mineral

composition, pore structure including its temporal dynamics, the biological activity in the soil food web and the dynamics of water and gas, to name just the most important factors. Moreover, all these different features relate in close interaction: water dynamics depend on soil structure, which is formed by soil biota, which itself depends on the structural soil properties with a feedback to vegetation and the quality of soil carbon . . . et cetera et cetera.

Hence, to predict the dynamics of soil functions in response to external forcing, the concept of empirical indicators needs

to be augmented by their dynamics at the timescale of the forcing under consideration. Here, the basic soil science disciplines can provide the required process understanding. Kibblewhite et al. (2008) criticize the "reductionist" approach of using simple indicators describing some fixed state. As a promising new approach, they suggest some form of "diagnostic tests" to directly evaluate the dynamics of soil in response to targeted forcing (e.g. compaction, added nutrients), so that the observable dynamics provide information on the internal pattern of interacting processes.

In the following, we suggest modeling the dynamics of soil as complex systems by identifying a larger set of "functional" soil characteristics and to analyze their site-specific dynamics and interactions based on both empirical observations and profound process understanding. Such an approach is expected not only to provide a dynamic component to the evaluation of soil functions, but also to identify crucial research needs for an improved understanding of the behavior of soil systems, their stability and resilience.

## 4  A new systemic approach for modeling soil functions

Starting from the insight that the dynamics of soils with respect to soil functions cannot be modeled based on first principles, an obvious question analogous to the search for suitable indicators is: what are the most relevant and observable soil properties that provide valuable information on soil functioning? Looking at the broad spectrum of observable soil attributes, as a first step we propose distinguishing three different attribute groups according to their characteristic time scale of change. There are

*inherent soil properties* that depend on the parent material and the stage of soil formation (e.g. the mineral composition, texture, layering, depth), which are not immediately affected by soil management and can be considered to be stable at a time scale of decades or more. In contrast, there are other observable soil attributes that may change at short time scales from minutes to days in response to external forcing (e.g. water content, temperature, redox potential, microbiological activity), which we refer

to as *soil state variables*. In between these extremes is a category, which we will refer to as *"functional soil characteristics"* that might change abruptly in response to external forcing, but have an intermediate time scale of change (days to months) as a result of internal processes and interactions. In this category are physical, chemical and biological characteristics as listed in Tab. 1, which is not intended to be comprehensive.

**Table 1.** Examples for soil functional characteristics reflecting a multitude of soil processes

| physical | chemical | biological |
|---|---|---|
| water capacity | organic matter | ecological engineers |
| macro-pores | pH | functional group diversity |
| aggregate stability | cation exchange cap. | ratio bacteria/fungi |
| . . . | . . . | . . . |

Based on the consideration of characteristic time scales of change, the category of functional soil characteristics is expected to carry the most valuable information on soil processes. The various indicators for soil functions that have been used in the literature (see above) is manifold, but all are in fact included in this category.

A major challenge is to identify a suitable set of functional characteristics and to derive meaningful indicators based on them, as illustrated in Fig. 2. There seems to be a consensus on which physical and chemical characteristics are essential in this category. This is still being developed for biological attributes. For instance, while most studies on effects of agricultural management measures on soil biodiversity have concentrated on taxonomic community parameters in the past, a more meaningful approach regarding soil functions would stress community functional diversity and/or key(stone) species driving specific processes such as bioturbation (Pulleman et al., 2012; Wall et al., 2010; Heemsbergen et al., 2004; Hedde et al., 2012). Generally, it should be noted that the impact of external forcing on functional characteristics may depend on the actual state variables (e.g. compaction due to traffic depends on the actual water content) and that the evaluation of derived indicators depends on inherent soil properties (e.g. relevance of macropores depends on soil texture).

Process-oriented research in the soil sciences mainly focuses on the dynamics of state variables (e.g. soil moisture, redox potential, soil respiration, soil biological communities), the related fluxes (e.g. evaporation, leaching of nutrients, greenhouse gas emission) and transformations (e.g. mineral N, biomass production, nutrient transfer). When modeling these processes, the category of functional soil characteristics is typically treated as static soil properties represented by suitable parameterizations (e.g. water retention curve, water capacity, hydraulic conductivity, porosity, pH, cation exchange capacity, biological composition or group abundances) as illustrated in Fig. 2. This seems to be too simplistic when changing the perspective towards the impact of soil management on soil functions. In this case, we need to account for the fact that the functional soil characteristics are also dynamic and affected by external forcing. Hence, it seems to be a formidable scientific challenge to extend the research on soil processes towards an in-depth exploration of the dynamics of functional soil characteristics and their interactions. Based on the understanding that these functional characteristics emerge from a multitude of process inter-

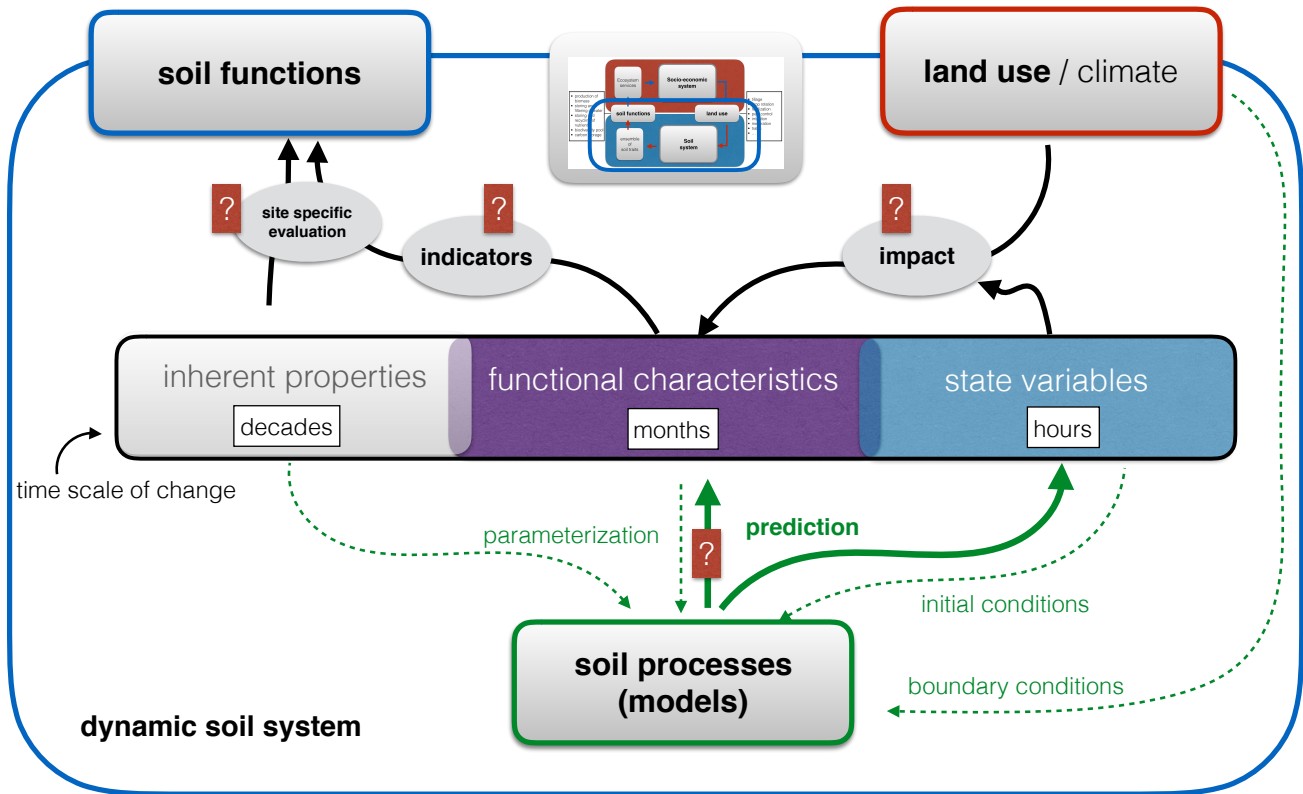

**Figure 2.** Zoom into Fig. 1 to illustrate the dynamics of soil functions within hierarchical categories of soil properties related to their characteristic time scale of change. Soil functions emerge from a combination of soil functional characteristics, which need to be evaluated according specific site conditions. Typical soil process models are focused on the dynamics of soil state variables, while soil functional characteristics are often parameterized and considered to be static. The actual challenges in predicting the impact of external forcing are marked by "?" (see text for further explanation).

actions at smaller scales, such a concept targets an intermediate (and hopefully manageable) level of complexity between the inextricably small-scale complexity and highly simplified static indicators. Nevertheless, the dynamics of state variables need to be included in the model concept since the impact of soil management or climate on functional soil characteristics depends on such state variables. For example, the change of soil bulk density due to external loads is highly sensitive to soil moisture
5  (see example below).

It has long been recognized that soils can be considered to be complex, self organized systems (Young and Crawford, 2004). However, a corresponding model approach is not yet available. Actually, the reaction of soils to external forcing exhibits some essential features which are typical for complex systems: within a certain range of forcing, soils are remarkably resilient against external perturbation, while beyond some critical point the state of soils may switch to some different mode or configuration.
10  Prominent examples are critical degrees of soil compaction, which can no longer be compensated by internal soil structure-

forming processes (Keller and Dexter, 2012) or some critical level of soil organic matter, below which soil degradation is invoked followed by a positive feedback loop: reduced OM → reduced biological activity → reduced nutrient cycling → reduced plant growth → reduced OM production . . . (Loveland and Webb, 2003).

For modeling the observed complex dynamics, we suggest focussing on the set of functional soil characteristics as introduced above. This provides a systemic perspective integrating the underlying complex process interactions. After nearly a century of quantitative pedology (Jenny, 1941), there is ample evidence that the state of such functional soil characteristics and especially their combination is not just random, but there are typical patterns related to soil types as defined by pedogenetic considerations. This is true for abiotic factors but is not so clear cut for biological species, where soil type alone often does not explain distributional patterns (Fromm et al., 1993; Lauber et al., 2008; Kanianska et al., 2016).

The US Soil Taxonomy (Soil Survey Staff, 1999) and the World Reference Base for Soil Resources (FAO, 2012) are explicitly based on the combination of such (abiotic) functional characteristics in addition to what we have identified as inherent properties (Fig. 2). The specific combination of soil functional characteristics that can be found at a specific location depends on the local conditions for soil formation and development, including parent material (i.e. geology), climate, topography, vegetation and land use. This has already been suggested by Jenny (1941) and still forms the basis for quantitative pedology today referred to e.g. in the SCORPAN approach (McBratney et al., 2003).

For the analysis of soil as complex systems, we suggest interpreting the traditional consideration of soil types as a characteristic combination of functional soil characteristics as illustrated in Fig. 3. Then, according to the terminology used for complex systems, a functional soil type is considered to be an attractor within the multidimensional state space of functional soil characteristics. An attractor is meant to be a combination of property states that are more frequently found than others, and the interpretation is that the underling soil processes and their interactions pull (i.e. attract) the system towards this state. This also implies that such attractors are relatively stable in response to external forcing, as is actually observed for soil.

An important corollary of this concept is that the set of functional characteristics is not a set of independent features, but the set members are all closely interrelated. This is evident in that all share the same basis of interacting soil processes from which they emerge. The type of interrelations as illustrated in Fig. 3 by springs are virtual and are expected to exhibit some elasticity. They are accessible not only through empirical observation, but might be derived from a profound process understanding, since they represent an integral manifestation of the underlying physical, chemical and biological processes.

Such "effective" relations between functional soil characteristics producing the postulated attractor are expected to be an essential key in describing the macroscopic behavior of soil in response to external perturbations. These relations are macroscopically observable, but the underlying processes that produce these relations are not easily accessible or measurable. They are expected to be interlaced and nested and, as typical for natural systems, highly non-linear.

Examples are the relation between SOM and aggregate stability, which is certainly not linear but expected to be an optimum function, or the relationship between burrowing macrofauna and soil bulk density, where the latter is obviously bounded by minimal and maximal values. We postulate that the stability and resilience of soils in response to perturbations can be described by the complex interplay between such effective non-linear relations. In the following section, we discuss the example of soil structure dynamics after compaction.

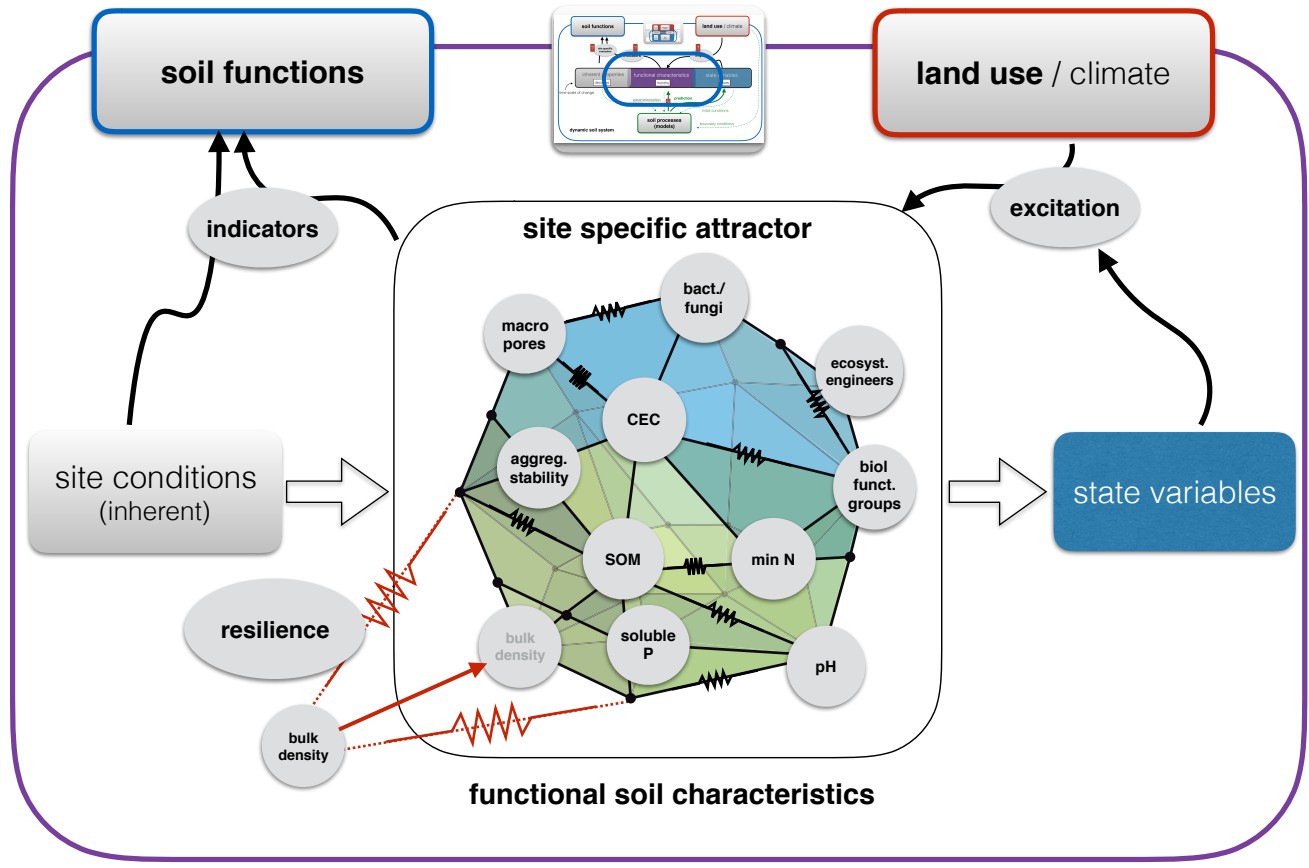

**Figure 3.** Zoom into Fig. 2: functional soil characteristics are related, forming a site-specific attractor as a quasi stable state. External perturbations (e.g. increase in bulk density by traffic) might be compensated by these relations or the system is pulled towards another stable state (see text for further explanation).

## 5 An example of systemic modeling of soil structure dynamics

It is well known that traffic on arable soil often leads to compaction and an increase in bulk density (illustrated in Fig. 3). This will impact the habitat pore space as well as the aeration and herewith the spatial distribution of redox conditions and the activities and interactions between soil organisms, which in turn affect nutrient availability for plant growth, et cetera. Hence, the functional soil characteristic abruptly affected first is the soil pore volume or soil bulk density. Other chemical and biological functional characteristics are then affected through subsequent feedback processes. While the physical impact of compaction as a function of load, soil texture and water content has been intensely studied (Hamza and Anderson, 2005; Keller and Auset, 2007) much less is known about the potential recovery of soil structure after compaction (Keller et al., 2017) as a result of structure (re)formation by plants, burrowing soil biota and physical processes such as swelling-shrinking and freezing-thawing. However, this is what we must know if we wish to evaluate the general impact of traffic and mechanical

loads on soil functions. In Fig. 4 some potentially relevant process interactions during soil structure recovery are illustrated. According to the proposed approach, the following processes should be considered and quantitatively described: the formation of pores by root growth, the supply of organic matter by plant roots, the stimulation of burrowing soil fauna by the available and supplied organic matter, the nutrient dynamics invoked by microbial activity stimulating plant growth, the impact of soil fauna on soil structure through bioturbation and the impact of swell-shrink dynamics on structure formation, just to name the most obvious. All these processes are intensely coupled since they all depend on the actual soil structure (and the related water and gas dynamics) while at the same time changing it.

The numerical coupling of all these interactions will be the next step. Our hypothesis is that this will lead to a system behavior developing towards some stable state after disturbance (i.e. compaction in this case). At some time scale, the system may recover as a result of small scale process interactions. If it does not recover, the internal interactions will draw the system to some other stable state, i.e. some degraded level. Therefore, the proposed modeling concept should also be able to identify critical tipping points in system behavior and, thus, critical thresholds with respect to external forcing.

Following the proposed concept, we need to identify the relevant soil functional characteristics and to connect them based on our current process understanding. There is substantial knowledge about individual processes, especially in the field of soil hydrology and soil carbon dynamics. It is yet limited for biological interactions. The overview required for such a systemic approach is still missing. This is especially true for interactions at the interface between different soil science disciplines and the interactions between physical, chemical and biological properties. For example, soil physics typically ignores chemical heterogeneities and biologically induced structure dynamics, while in biology and chemistry soil analyses are often performed in homogenized or standardized samples and the natural structure/habitat is lost (Heemsbergen et al., 2004; Crowther et al., 2012).

The proposed concept will also reveal new research questions which are essential for understanding the system's behavior. Following the example of soil compaction and relaxation (Fig. 4), such a missing link is, for example, the impact of root growth on soil structure development. More specifically, we need to know the affinity of plant roots to grow into existing pores or their capacity to generate new pores. We expect this to be a function of soil texture, bulk density, soil moisture and certainly depends on the plant species as well. To the best of our knowledge, there is very little experimental evidence currently available along these lines, although much progress has recently been made to investigate plant roots by non-invasive imaging (Downie et al., 2015; Mooney et al., 2012).

Obviously, our process understanding will always be limited and some of the implemented interactions might be pure speculation. Nonetheless, the proposed modeling framework will provide a valuable tool for evaluating alternative process descriptions with respect to their impact, sensitivity and importance for the system's behavior. A promising perspective is that the uncertainty in the assumed process interactions is expected to decrease with ongoing research.

This provokes the question of how to eventually validate such a model approach. In fact, the model behavior needs to be assessed with historical and newly generated observations to evaluate its plausibility and to develop new insights for the formulation of site-specific process interactions. A crucial problem is the longer time scales in which soil processes rebound or move to an alternative state and their adaption to changing boundary conditions. Therefore, another valuable source of

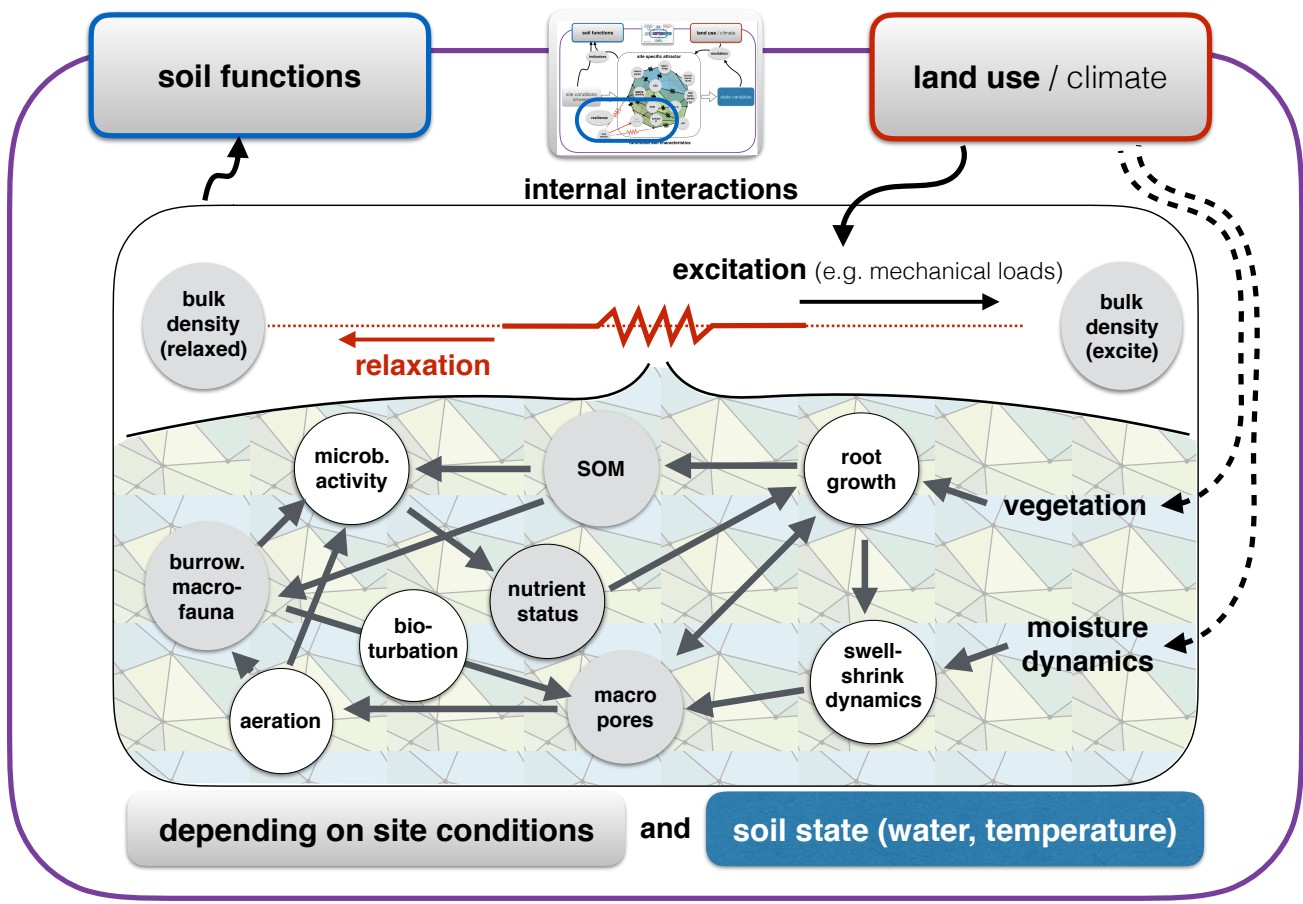

**Figure 4.** Zoom into Fig. 3: The excitation (i.e. disturbance) of the soil system by compaction due to mechanical loads is relaxed by interacting processes at the smaller scale. The functional characteristics considered for this case are indicated by grey circles while the connecting processes are in white.

information are long-term field experiments where the history of these boundary conditions is well documented. Yet another option is to involve farmers who may provide the historic information that led to the actual state of soil functional characteristics in their fields.

## 6 Conclusions

5 We propose a concept for a systemic approach to modelling soil functions and their dynamics. All detailed process research being carried out in the soil sciences can substantially contribute to improving the scientific fundament of this approach, which is especially true for the exploration of interacting processes leading to stable configurations of the soil "functional characteris-

tics". The set of these functional characteristics and the level of complexity can be adapted to specific soil functions of interest and developed according to the growing state of knowledge. Hence, the proposed modeling framework may continuously grow with respect to scientific evidence. In other words, it paves the way from simple rules of farmers' proverbs to sophisticated scientific analyses.

5    Starting from the pressing need to predict the impact of soil management measures on essential soil functions, we developed a systemic modeling framework based on the complex interactions of physical, chemical and biological processes. It forms a basis to use past and ongoing soil research for evaluating soil functions. Thereby, not only the actual state but also the dynamics of soil functions in response to external forcing induced by land use and climate can be predicted. We consider this to be of utmost importance for making decisions on soil management options in the light of sustainability.

10  *Acknowledgements.* This research has been funded by the German Ministry of Education and Research (BMBF) in the framework of the funding measure "Soil as a Sustainable Resource for the Bioeconomy - BonaRes" (grant 031A608). For further information please visit www.bonares.de.

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
