# Peer review of "A systemic approach for modeling soil functions"

_SOIL, 2017_

## Referee Comment (RC1) · Prof.Dr.Ir Bouma (Referee) · 9 Oct 2017

Sustainable soil management requires a systems approach (SOIL-2017-26) Vogl et al

Review comments J.Bouma.

General comments In a well written paper, the authors present a relevant question: " what can the contribution of soil sciences be to sustainable soil management"(p2, line6). They later ( p5, lines 4-5) specify a goal:"the ultimate goal of soil research's support for sustainable soil management is to quantify and predict the impact of external forcing on the ensemble of soil functions". And they mention the means to achieve all this:'modeling soil as a complex, adaptive system"(p3, lines4-5). But they also indicate that we have a long way to go:"to truly capture the reaction of soils to external forcing

through land use and climate change ..is still in its infancy"(p2, lines 17-18). Still, "soil scientists are working on detailed process understanding (p4, line 9). Perhaps these, as such correct, statements could be put together in a unified storyline. This type of paper is valuable, in my view, particularly at this point in time when soil science has spectacular opportunities as UN Sustainable Development Goals have been approved by 195 states, members of the UN, in 2015. This includes legal obligations to report progress towards 2030. Indeed, as has been observed elsewhere, in soil science sub-disciplines still operate rather independantly. I fully agree with the statements on p2, lines 17-18 and p10, lines 24-27. How to cope with this is a highly relevant question. I have a number of questions, comments and suggestions that will be articulated below. I will cite a number of publications with the only objective to better document comments made. I am aware of the recent "citation- stacking' excitement that does, of course, not apply in this context (and that is irrelevant anyway in my case because of my age). Three overarching comments: (1) the paper is strongly soil-focused, while I feel that now more inter- and transdisciplinarity is also needed, emphasizing the word:"also", because we have to continuously develop our own science; (2) the schemes are very complex and it would have been useful to include some specific examples to increase transparency, (3) the authors correctly conclude that modelling the complete soil system with all its interconnections is still not possible but should be pursued in future. Then they jump rather abruptly to defining indicators. I feel there is a way in between, to be further explored below.

Specific comments 1. P2, line 14: the main message of Keesstra et al ( 2016) and Bouma and Montanarella ( 2016) was the need to link soil functions with ecosystem services that are, in turn, interpreted to define ways to reach SDGs. Soil scientists can't do it alone, addressing the SDGs that have a strong and broad societal focus when the corresponding texts are read completely. We, and many others, have found that using dynamic simulation models of the soil-water-plant-climate system ( e.g SWAP but many other models as well) is a highly effective and functional way to realize interdisciplinary interaction. The word modeling is used a lot in this paper, but what does it mean? I

suggest the soil-water-plant-atmosphere modeling approach. We have lots of soil data in databases and we have to be aware of the manner in which such data are used by agronomists, hydrologists, climatologists and ecologists. There is a tendency to pick up texture, bulk density and %C, use pedotransferfunctions and assume that the soil part of soil-water-plant–climate models is being covered ( and input from soil scientists is not needed anymore). This way "soil" is all too often represented very poorly. So we have to stay involved in the modeling process, asking how soils can be represented best. The questions raised in this particular SOIL paper address this very issue.

2. The authors address the important issue how current land-use questions are being identified ( p2,line19). I would be in favor of pro-active approaches by (soil) scientists, engaging stakeholders. This is further explored in two publications that were published after this SOIL paper was written. I invite the authors to consider these observations: Bouma, J., 2017. How Alexander von Humboldt's life story can inspire innovative soil research in developing countries. SOIL 3, 153-159. (https://doi.org/10.5194/soil-3-153-2017). Bouma, J., 2018. The challenge of soil science meeting society's demands in a "post-truth", "fact- free"world. Geoderma 310, 22-28.(http://dx.doi.org/10.1016/geoderma2017.09.017).

3. The "major societal concerns"( Baveye, 2015) (p2, line 23) are now well expressed by the SDGs. I would avoid the terms "services that soils provide"( p2, line 25). For clarity: soil functions contribute to ecosystem services. The "services"mentioned on p2, line 25, are soil functions ( see also p3, line 26: indeed, soil functions are not the same as ecosystem services). I like the reference to the DPSIR approach (p2, line 16) but I interpret the system somewhat differently in terms of pro-actively offering options to stakeholders and policy makers, from which to choose ( Bouma, 2018).

4. I wonder whether the link between social and natural sciences is clear but I certainly agree that the interface needs to be further developed. I refer again to the two papers cited above. We need, I feel, to pay particular attention to developing countries to encourage them to develop their own independant approach focusing on basic principles

involved.

5. The authors take a big jump when moving from modeling interactive soil physical, chemical and biological processes ( which is indeed very difficult) to indicators, to be discussed later. There is, in my view, an intermediate possibility. When we identify a given soil, we accept it as it presents itself and we describe it and measure its properties. There have been some studies that try to model soil formation, starting with the unchanged parent material and covering often thousands of years. Very difficult as is modeling of interacting soil processes associated with land use, as discussed in this paper. So why not look at the effects of different forms of land use as it presents itself in the field as a function of management, that can be traced back by questioning farmers. Sonneveld and Pullemans did so for SOC, each looking at a particular soil type and measuring soil properties at 50 locations, identified by using the soil map, while identifying past land use. They could develop regression equations that predicted %C remarkably well as a function of past and current land use:

Pulleman, M.M., J. Bouma, E.A. van Essen and E.W. Meijles. 2000. Soil organic matter content as a function of different land use history. Soil Sci. Soc. Amer.J:64,689-694 Sonneveld, M.P.W, J.Bouma and A.Veldkamp. 2002. Refining soil survey information for a Dutch soil series using land use history.. Soil Use and Manag.18:157-163.

So:go back to the field and observe soils that have been subjected to particular forms of land use. Establish effects by measurements and this way obtain a characteristic range of properties as a function of land use for any given soil type ( even thoughunderlying interacting processes are unknown in detail but often in a general way).

6. Indicators are indeed important (p5, line 14). Soil health is mentioned here in passing but this needs more attention as the topic is quite "hot"in the USA. Recent developments ( Moebius-Cloene etal 2017) define, aside from the traditional chemical indicators , also soil physical ones and, dominantly, soil biological ones. These authors also feel that soil characterization has focused too much on chemistry in the past. They

define numbers for physical, chemical and biological soil health and put them together but in a rather unclear manner, that certainly does not represent the "dynamic"approach that Kibbleworth etal2008 mention. :

Moebius-Cloene et al, 2017. Comprehensive assessment of soil health. The Cornell Framework. (http://www.scs.cals.cornell.edu) . See also: www.soilhealthinstitute.org

7. I fully support the introduction of "threshold"values, also called "tipping points". They have successfully been defined for ecosystems and are quite relevant for soils. I can't resist to quote the following paper, introducing pedotransferfunctions ( later –pedo was added) and threshold values:

Bouma, J. and H.A.J. van Lanen, 1987. Transfer functions and threshold values: from soil characteristics to land qualiÂñties. In: Quantified Land Evaluation. Proc. of a workshop by ISSS/SSSA. ITC-Publication no. 6. p. 106-111.

Indeed, identify tipping points by process studies but, I would suggest, also by field observations! Yes, the authors are quite correct that the dynamics of soil functions needs to be determined, but that can also be done by making multiple field observations at critical points in time. Yes, when is compaction so severe ( threshold bulk density) that roots cannot penetrate the plowlayer and are there sites where farmers have deep-plowed and seeded deeprooting plants etc. The authors mention that a deep understanding of the underlying processes is needed ( p5, line 33) but this a hard call because compaction and sheer forces interact in a quite complicated manner. So, keep investigating the processes, certainly, but also observe the effects defining thresholds , most probably of critical water contents, corresponding to the lower plastic limit ( lacking a better measure than this one from 1915). A specific example:

Droogers, P., A. Fermont and J. Bouma. 1996. Effects of ecological soil management on workability and trafficability of a loamy soil in the Netherlands. Geoderma 73: 131-145..

Also, indeed, what is a critical SOC value etc. Resilience is briefly mentioned (p7, line 18). Important as well. Some soils are more resilient than others! I still remember well young volcanic soils in Costa Rica (Andisols) that recovered rapidly from compaction after deforestation, while old volcanic soils ( Ultisols) did not:

Spaans, E., J. Bouma, A. Lansu and W.G. Wielemaker, 1990. Measuring soil hydraulic properties after learing of tropical rain forest in a Costa Rican soil. Tropical Agriculture 67: 61-65.

8. Soil types and soil classification are mentioned on page 8, lines 106. That is too late in my view. I strongly believe that soil types ( soil series in the US) have a characteristic "story to tell" and that stratification by soil type is meaningful, if not essential, in creating a systematic approach. In an attempt to express the ( characteristic) range of properties of a given soil type as a function of different forms of management , we defined genoforms ( the classification name based on what you call inherent soil properties) and phenoforms that express the effects of management ( your functional characteristics):

Droogers, P. and J. Bouma. 1997. Soil survey input in exploratory modeling of sustainable soil management practices. Soc. Amer. J. 61: 1704-1710.

Conclusion: the discussion being initiated by the authors is relevant for soil science at this crucial point in time. I suggest that the author consider comments made above to broaden the scope of their analysis.

Please also note the supplement to this comment:
https://www.soil-discuss.net/soil-2017-26/soil-2017-26-RC1-supplement.pdf

---

## Referee Comment (RC2) · Anonymous Referee #2 · 7 Jan 2018

Sustainable soil management requires a systemic approach

Vogel et al.

This is a well written paper, albeit somewhat repetitive at places, and it puts forward an interesting and novel approach to modelling soil systems. Even though it is somewhat abstract at this stage it makes valuable points that can add to the debate how soils should be modelled in a meaningful way. I am not convinced the title captures correctly the novelty introduced in the paper; the novelty is not that a systems approach is required (this has been made numerous time before), but the novelty centres on the approach they put forward so I would encourage the authors to search for another title that better captures the essence of the paper. The proposed modelling concept is very abstract as it is currently presented. I suspect the authors have already explored some

of the behaviour of such an approach and if so it might add value if they could make the approach more accessible by including an example. If that isn't possible at this stage, perhaps they could include a discussion on how this approach could be used in 'real' situations? I can see how the functional characteristics can be quantified, but I find a less obvious how the authors have decided what important functional characteristics are (and what are not); the fact that table 1 only list a few (apparently on purpose at this stage) makes me wonder how they envisage this to be progressed to modelling. Secondly, the behaviour of the system they project in Fig 3 is most likely to be determined by the interactions between the 'functional characteristics'; the authors have given this relative little consideration in their paper and I would welcome insights in how these could be measured (if possible et al), or what other means the authors consider to parameterise the system. This makes it difficult to assess if this is a viable option that is being put forward or if it is an alternative approach that retains difficulties they identified in other approaches. The above questions do not devaluate the approach they put forward in this paper; I believe that the challenging of current approaches is valuable and the gap in our knowledge appears correctly identified in this paper, whatever modelling approach we consider. Some additional comments are given below: P2 line 7: reminder $\Rightarrow$ remainder P2 line 10-18: I am not sure why the authors conclude that based on these papers the impression may e reached that our scientific knowledge on soil processes. . .. Is pretty much settled. . ..' I did not reach that conclusion from those papers, and it seems to me that this sentence was merely included to introduce to emphasise their view that this is a misimpression. Fig 1 is rather simplistic; this may be essential to introduce the remainder of the paper, but it lacks, for example, feedbacks with the climate, suggesting soil management being the only factor impacting upon soil. In going forward it is essential that our systems are sustainably management against the backdrop of a changing climate. I am not advocating expanding the figure, but a additional discussion of the boundaries of the system they consider might be welcome; it currently appears very centred on soil. It may also need to be recognised that there is a role for land management and not just soil management. Arguably the examples

they give in the figure aren't 'soil management' in the strictest sense, but agronomic measures. It would help in the caption of Fig 1 to make reference of details being described in Fig 2 and 3; some of the initial concerns (lack of feedbacks) I noted were resolved later in the article. P5 line 7: the 'interactions between physical, chemical and biological processes are mentioned repeatedly in the paper. P5 line 13: I agree with the limitations put forward against the bottom up approach, but would have welcomed a discussion against these points for the approach they put forward: is it really better? Is the spatial heterogeneity less of a problem? Is the lack of information on interactions not a problem? Is there no non-linearity in your system and/or interactions? It is unclear to me if the approach you put forward is 'different' or better in this respect. P6 line 30: you state that functional soil characteristics carry most valuable information; most valuable for what?, upon what is this statement based? How did you come to the set of functional characteristics in table 1? E.g. you list macro-pores which indeed can be a longer term process but at the same time changes almost instantaneously in response to wheel traffic/compaction. Fig 2: it may not be possible to link state variables to soil functions or ecosystem services in an empirical way, due to the dynamic nature of these, but progress has certainly been made in modelling these processes and relating these model outcomes to soil functions, ecosystem services and how these may respond to management. As such the focus on the state variables may not be as futile as is suggested by the authors as it fulfils a valuable role in model validations. I agree however with the authors that there is a gap in our knowledge on what they identified as functional characteristics. However, as indicated in their fig 2, the focus on functional characteristics leaves us with uncertainty about impact and indicators; they therefor appear to have replaced one set of uncertainties and unknowns hampering modelling with another. As indicated above some broader discussion on how this approach could be implemented might strengthen their call for an alternative approach.

P10 line 28: what is the new research you are referring too here? Surely soil-root interactions is not a 'new area of research'? You overlook the multitude of research in this field including the progress with modern technologies; it would be more helpful,

if you wish to keep this argument, to be more specific about what it is precisely that you require to progress your modelling. It seems to me that the interactions between the functional characteristics are a new area of research. This in my view is generic to systems approach that the interactions are critical to its behaviour, yet often unknown. I think if the authors could consider some of the above comments in their revision it would be stronger as a call for a different approach. I think it can make a valuable contribution to the academic community and stimulate debate on how best to model soil systems.

---

## Editor Comment (EC1) · P.D. Hallett (Editor) · 17 Jan 2018

Thank you for your robust response to the 2 reviewers. We look forward to receiving the revised manuscript that has addressed the comments as described.

---

## Author Comment (AC1) · 17 Jan 2018

We thank Johan Bouma for his valuable comments which we are glad to include in our paper. Here some comments on how we intend to do this.

General comments:

1) Clear Storyline: We will summarize the major goals, related problems and proposed avenue to proceed into a more coherent story line at the end of the introduction.

2) The paper is strongly soil-focused, while more inter- and transdisciplinarity is also needed: We definitely agree and made this point in the introduction - but it is fully intentional to focus this paper on what actual science can contribute. We feel that this question was actually somewhat underexposed in the current discussion on soil

functions, ecosystem services and SDGs. Yet, we emphasize that a transdisciplinary approach is highly required for sustainable soil management. Another paper is actually in review where the focus is on the transdisciplinary aspect (Helming et al: Managing soil functions for a sustainable bioeconomy – assessment framework and state of the art, submitted to LDD)

3) Include some specific examples to increase transparency: We will extend the presentation of this case study in an additional section (2.3) to better illustrate how the model can be applied and what input is actually required.

4) There should be a way in between modeling soil in full complexity and using simple indicators: We agree - but in fact the proposed focus on "functional characteristics" and their interactions is meant to be such an intermediate approach. We will make this more clear in section 2.2.

Specific comments:

1) We absolutely agree that dynamic simulation models are very important also for stakeholder interaction and just want to stress the special challenge for (natural) soil science in this framework (see general comment 2). We of course also agree that the representation of soil in soil-water-plant-atmosphere models is typically too simple and will emphasize this more clearly. In fact this justifies the more elaborate model concept we suggest here.

2) Pro-active approaches by soil scientist to engage stakeholders: This is a very important point and we will emphasize the potential of process-based soil models (including the one which is presented here) to demonstrate the sensitivity of soil functions with respect to external perturbation. In this context we are happy to include the suggested references.

3) We agree and change the wording from "services that soil provide" to " the contribution of soil functions to ecosystem services "

4) The link between social and natural sciences: Fig.1 is rather simplistic but we hope it describes the two essential interfaces between the societal and the natural systems which should in principle be pretty general and valid also for developing countries. We will extend the explanations for Fig.1 as also asked for by reviewer #2.

5) The big jump from complex process interactions to simple indicators: In section 2.1. we contrast the two extreme approaches for evaluating soil functions: Modeling complex process interactions starting from the molecular scale and just using simple indicators. As a conclusion we actually suggest an intermediate possibility: identifying these "functional characteristics" providing integral information on soil processes and attempting to model their dynamic interactions in response to external forcing. The suggestion to profit from some known history of soil management obtained by questioning farmers is very good! This is especially valuable as an addition to analyzing long term field experiments - so we will add this idea to section 2.2 as an additional option to verify and further develop the proposed model approach.

6) We will emphasize the concept of soil health and will add the reference to Moebius-Clune et al. 2017

7) We agree that field observations are absolutely essential to identify process interactions and especially tipping points (or threshold values). This is why the model development and the formulation of the required interactions need to be based on exactly such observations and focused experimental studies. This is our motivation to set up a structured library for published observations as another activity of the BonaRes project - this will be subject of a forthcoming paper. Here we will add the suggested idea to look for cooperating farmers who are willing to share the history of soil management which can be evaluated and connected to the present state of soil properties as an additional observational tool. We also agree - and actually tried to make this point very clear - that the assessment of soil functions and herewith also tipping points need to be site specific.
8) We agree that the importance of soil types needs to be mentioned earlier and will do so already in the introduction (where it has been mentioned only in passing).

---

## Author Comment (AC2) · 17 Jan 2018

We thank the reviewer for his/her critical but very constructive and valuable comments! We will follow the general suggestion and reduce the obvious redundancy at several places. We will change the title to "A systemic approach for modeling soil functions". In the following we discuss the specific points brought up in the review:

1) Specific example for application of the model required: Indeed, in the meantime we developed the model and especially the case study on compaction and recovery of soil structure further. We will extend the presentation of this case study in an additional section (2.3) to better illustrate how the model can be applied and what input is actually required. However, the numerical implementation will be part of a separate paper since

this would blow up the paper enormously and the special focus on the underlying model concept might be diluted.

2) How do we identify functional characteristics? The starting question is: which measurable/observable properties are carrying substantial information on soil functions and provide integral information about small-scale processes and their interactions? In fact this is the question how to upscale soil process descriptions. It came out that the possible candidates of such properties coincide with properties that change at an intermediate time scale of months to years in soils (of course macro pore volume may change abruptly below a tractor wheel, but the dynamic change under natural boundary conditions and the recovery after compaction is at a much longer time scale). Moreover, what we ended up with in Tab.1 (which of course might be extended) is in accordance to a considerable body of recent literature on indicators for soil functions and own analysis published in recent review papers (Rabot et al. 2018, Wiesmeier et al. 2018). We have the impression that we are converging towards some limited number of suitable "functional characteristics" in our modeling framework. These functional characteristics may eventually translate to indicators of soil functions.

3) How to include functional characteristics into modeling? They need to be a subset of the interacting "agents" in the process interaction network. In fact they are slowly changing state variables in our model concept. This is in contrats to many other process models in soil where these characteristics are treated as fixed material properties. We will illustrate this by highlighting the functional characteristics on the nodes of the network for the specific example in Fig. 4.

4) How to measure or parameterize the process interactions? Indeed most of them cannot be directly measured but for many of these interactions there is substantial site-specific knowledge or lets say a reasonable hypothesis about the principle form of these interactions in the literature. There are other process interactions for which much less is known about. And we need to rely on plausible assumptions (Examples are provided along with the example in Fig. 4). An attractive perspective is that process

interactions can be updated as soon as new insight is generated. Also the model concept can be used as a "toy model" in sense of: what would be the consequences if the interaction changes from type A to B ?

5) Where does the impression comes from that our scientific knowledge is often thought to be pretty much settled? This impression is not only based on the cited papers which are focused on the need for transdisciplinarity (certainly very important) and, thus, less on requirements in the field of natural sciences. It is also based on other, often non-scientific publications in this field and events like the "soil water nexus" and the "global soil week" which are mainly dedicated to the question how to bring the available knowledge into practical decision support (again, also important). Yet, we feel the need to make a strong point that fundamental soil science is still required and how this could provide a substantial contribution.

6) The simplistic Fig.1 is motivated to illustrate the human-soil interface and since there is no direct societal impact on climate it is missing here. We will make a strong point in the text that from the perspective of soil systems, climate is the other essential driver besides agricultural soil management. In Fig. 2, where the focus is on the soil system, climate has already been included. We will extend the caption of Fig. 1 to refer to the more detailed Figs 2 and 3 as suggested.

7) Why do we believe that our approach is superior to a bottom up approach starting at the molecular scale? We will add a paragraph at the end of section 2.2 to summarize why we focus on observable "functional characteristics" and where to get the information about the relevant interactions which of course can be non-linear as well. The problem of spatial heterogeneity is reduced since the model approach is focused on entire soil profiles and a stratification of functional characteristics considered to be "effective" descriptions of soil horizons. When it comes to larger scales we face the same classical problem: how are different soils distributed in the landscape - but this is not the scope of this paper.

8) The question how do we come to the functional characteristics has already been discussed above (2).

9) What is the importance of state variables? We definitely agree that modeling the dynamics of state variables (e.g. water content) provide highly valuable information on soil functions (e.g. water storage) - but to evaluate the capacity of a soil to store water we rather focus on the bulk density (or inferred hydraulic properties) and not on the actual water content. As indicated in Fig. 2 state variables are also highly relevant for the impact of external drivers on the functional characteristics. For example, the impact of wheel traffic on water capacity is very sensitive to the actual water content. This is why the dynamics of state variables need also be part of the proposed model concept. We will make this point more clear.

10) The focus on functional characteristics leaves us with uncertainties about impact and indicators: For sure there are considerable uncertainties at various places. However the focus on functional characteristics allows us to base the model approach on observable properties having their individual but in principle measurable variability. The uncertainty related to the impact is included in the process interactions formulated as the kernel of the model approach. Here the perspective is that the uncertainty will decrease with increasing process understanding. The indicators are thought to be a subset of the functional characteristics or derived as a combination of such a subset depending on the soil function to be evaluated. Hence their uncertainty is directly related to the uncertainty in the observation of functional characteristics. This discussion will be included in the additional paragraph (see 7).

11) What is the new research we are referring to? We agree that a lot of progress has been made in the field of plant soil interaction especially with the help of new visualization techniques (and this is very promising to support our model approach). For the specific problem described in the paper – the recovery of soil structure from compaction – the specific point we are referring to is that we need to know the affinity of plant roots to grow into existing pores or their capacity to generate new pores. We

expect this is a function of soil texture, bulk density, soil moisture (again dependent on state variables!) and it depends on the plant species as well. To the best of our knowledge there is very little experimental evidence currently available along these lines. We think that this relatively new types of questions will pop up if we try to formulate the relevant process network required to predict the dynamics of functional characteristics in response to various drivers – we think this is an exciting endeavor and will include this discussion in the more detailed description of the example (see 1).